# Phosphorene/rhenium disulfide heterojunction-based negative differential resistance device for multi-valued logic

Jaewoo Shim[1], Seyong Oh[1], Dong-Ho Kang[1], Seo-Hyeon Jo[1], Muhammad Hasnain Ali[1], Woo-Young Choi[1], Keun Heo[2], Jaeho Jeon[3], Sungjoo Lee[3], Minwoo Kim[3], Young Jae Song[3] & Jin-Hong Park[1]

Recently, negative differential resistance devices have attracted considerable attention due to their folded current–voltage characteristic, which presents multiple threshold voltage values. Because of this remarkable property, studies associated with the negative differential resistance devices have been explored for realizing multi-valued logic applications. Here we demonstrate a negative differential resistance device based on a phosphorene/rhenium disulfide (BP/ReS$_2$) heterojunction that is formed by type-III broken-gap band alignment, showing high peak-to-valley current ratio values of 4.2 and 6.9 at room temperature and 180 K, respectively. Also, the carrier transport mechanism of the BP/ReS$_2$ negative differential resistance device is investigated in detail by analysing the tunnelling and diffusion currents at various temperatures with the proposed analytic negative differential resistance device model. Finally, we demonstrate a ternary inverter as a multi-valued logic application. This study of a two-dimensional material heterojunction is a step forward toward future multi-valued logic device research.

[1] School of Electronic and Electrical Engineering, Sungkyunkwan University, Suwon 440-746, Korea. [2] Frontier Technology Lab, R&D Headquarters, SK Hynix Co. Ltd., Ichon 460-701, Korea. [3] Sungkyunkwan University Advanced Institute of Nanotechnology, Sungkyunkwan University, Suwon 440-746, Korea. Correspondence and requests for materials should be addressed to J.-H.P. (email: jhpark9@skku.edu).

Recently, negative differential resistance (NDR) devices have attracted considerable attention owing to their folded current–voltage (I–V) characteristic (N-shaped I–V curve), which presents multiple threshold voltage values[1–27]. Because of this remarkable property, studies associated with the NDR devices have been explored for realizing multi-valued logic (MVL) applications[1,7,11,13,26]. Compared to conventional binary logic systems, MVL systems can transmit more information with fewer interconnect lines between devices by transferring multi-valued signals, thereby reducing the complexity of modern integrated circuit design. For example, a ternary logic system is theoretically able to reduce the number of interconnect lines by nearly 45% as compared with binary logic[28]. The NDR devices that have been researched for the implementation of this MVL system are Esaki diodes[2–7], resonant tunnelling diodes[8–20], Gunn diodes, single electron transistors[21,22] and molecular devices[23,24]. However, at the current stage of research, because most of the Esaki diodes and resonant tunnelling diodes were fabricated in Si–Ge and III–V semiconductors[2–4,8–14], the formation of various types of heterojunctions (type-I, II and III) is limited by threading dislocations, which are normally caused at the junction interface by lattice mismatch during film growth. Although the threading dislocation that increases the valley current of NDR device can be reduced by applying superlattice and nanowire structures, it is hard to avoid that the fabrication process becomes more complex.

In this light, atomically thin two-dimensional (2D) semiconductors, such as molybdenum disulfide ($MoS_2$), tungsten diselenide ($WSe_2$), rhenium disulfide ($ReS_2$), tin diselenide ($SnSe_2$) and black phosphorus (BP), are expected to offer attractive material platforms for NDR devices due to the absence of dangling bonds on their surfaces. Because these 2D semiconductor layers are stacked via weak van der Waals interaction, 2D materials-based heterojunctions do not suffer from lattice mismatch and form atomically sharp interfaces, allowing high-quality heterojunction interfaces[29–31]. It is also possible to design various heterojunctions by stacking different 2D materials with different bandgaps and electron affinities, where band structure alignment can be classified into three types: type-I (straddling gap)[32], type-II (staggered gap)[2–4,6,7,33–36] and type-III (broken gap)[5,32]. Recently, Roy et al.[6] reported an NDR device based on an $MoS_2/WSe_2$ heterostructure, which was fabricated in a type-II heterojunction. However, the NDR device used dual gates involving a complicated fabrication process to obtain an electrostatically doped $n^+/p^+$ heterojunction, and the NDR behaviour was observed at a very low temperature below 175 K. Nourbakhsh et al. and Yan et al. also reported NDR devices in $MoS_2/WSe_2$ and $BP/SnSe_2$ heterojunctions, respectively[5,7]. In these devices, it was necessary to use the specific thickness of 2D semiconductors to ensure band-to-band tunnelling of carriers, and the obtained peak-to-valley current ratio (PVCR) values were lower than 2 at room temperature.

Here, we demonstrate an NDR device based on a $BP/ReS_2$ heterojunction that is formed by type-III broken-gap band alignment, showing high PVCR values of 4.2 and 6.9 at room temperature and 180 K, respectively. In addition, as an MVL application, we present a ternary inverter (having three states) that combines a $BP/ReS_2$ heterojunction NDR device and a BP p-channel thin film transistor (TFT). This integration approach based on NDR devices is expected to fulfil low-power advantages of future MVL circuits by reducing the parasitic interconnect capacitance. In particular, compared with a type-II heterojunction, a type-III heterojunction can easily implement a highly doped $n^+/p^+$ heterojunction without a separate process, such as electrostatic doping by gate bias and a chemical doping process. First, we confirm the broken-gap band alignment of the $BP/ReS_2$

heterojunction with Kelvin probe force microscopy (KPFM). Then, the carrier transport mechanism of the $BP/ReS_2$ heterojunction NDR device is discussed in detail at room temperature. Furthermore, through temperature-dependent current–voltage (I–V) measurements and the proposed analytic NDR device model, where tunnelling/diffusion currents and parasitic series resistance were considered simultaneously, we quantitatively study the temperature-dependent device operations.

## Results

**Characteristics of BP/ReS_2 heterostructure.** Figure 1a presents schematic diagrams of the $BP/ReS_2$ heterostructure on an $SiO_2/Si$ substrate. A BP flake was prepared onto an $SiO_2/Si$ substrate by a tape-based exfoliation method[37], and then an $ReS_2$ flake was transferred onto the BP flake by a mechanical transfer process method (the optical images of the $BP/ReS_2$ heterostructure can be found in Supplementary Fig. 1)[38]. The thicknesses of the BP and $ReS_2$ flakes were confirmed by atomic force microscope to be about 40 and 50 nm, respectively (Fig. 1b,c). Figure 1d shows the Raman spectra obtained at three different positions in the $BP/ReS_2$ heterostructure sample, where the spectra from top to bottom indicate a $ReS_2$ region, a $BP/ReS_2$ overlapped region and a BP region. The observed Raman peaks of BP at 366, 442 and 470 $cm^{-1}$ correspond to the $A^1_g$, $B_{2g}$, and $A^2_g$ phonon modes, respectively. This Raman spectrum for $ReS_2$ includes two prominent peaks at 154 and 215 $cm^{-1}$, which are attributed to the in-plane ($E_{2g}$) and out-of-plane ($A_{1g}$) vibrational modes. The Raman spectrum of the overlapped $BP/ReS_2$ region contains the vibration modes of both BP and $ReS_2$, indicating the formation of a heterostructure. Next, to investigate the band alignment of the $BP/ReS_2$ heterojunction, we carried out KPFM measurements. Figure 1e shows the three-dimensional KPFM mapping image of the $BP/ReS_2$ heterostructure and contact potential difference ($\Delta V_{CPD}$) histograms extracted from the mapping image. Before the KPFM measurement, the KPFM tip (platinum/iridium (Pt/Ir)-coated Si tip) was calibrated on a highly oriented pyrolytic graphite (HOPG) surface. Here, the HOPG is conventionally used to calibrate the work function of the KPFM tip because it has a clean surface and its work function is well known to be 4.6 eV (ref. 39). The average $\Delta V_{CPD}$ values on the BP and $ReS_2$ flakes were obtained at $-153$ and 430 mV, respectively. Since the $\Delta V_{CPD}$ is the difference in the work function between the KPFM tip and the sample (inset of Fig. 1f), the work function values of the BP and $ReS_2$ can be calculated using the following equation: $\Phi_s = \Phi_{tip} - \Delta V_{CPD}$, where $\Phi_s$ and $\Phi_{tip}$ are the work functions of the samples (BP and $ReS_2$) and the KPFM tip, respectively. Here, $\Phi_{tip}$ was obtained from the sum of the HOPG work function ($\Phi_{HOPG}$) and $\Delta V_{CPD}$ between the KPFM tip and the HOPG surface ($\Phi_{tip} = \Phi_{HOPG} + \Delta V_{CPD\_HOPG}$), which is presented in more detail in Supplementary Fig. 2 (refs 40,41). Therefore, the work function values of the BP and $ReS_2$ films can be estimated to be about 4.5 and 5.1 eV, respectively (Fig. 1f). Based on the obtained KPFM results and the previously reported band properties (conduction band minimum, valence band maximum and band gap ($E_g$)) of BP and $ReS_2$ (refs 42–44), we graphically described the predicted energy band alignment of the BP and $ReS_2$ heterojunction at equilibrium before contact (Fig. 1g) and after contact (Fig. 1h). Here, the conduction band minimum, valence band maximum and $E_g$ values of the BP ($ReS_2$) that were calculated using a first-principles density of states in the literature were 4.2 eV (4.68 eV), 4.59 eV (6.05 eV) and 0.39 eV (1.37 eV), respectively. As shown in Fig. 1g, the $BP/ReS_2$ heterojunction seems to form a broken-gap band alignment (type-III heterojunction) because the highest valence band edge of BP is located above the lowest conduction band edge of $ReS_2$.

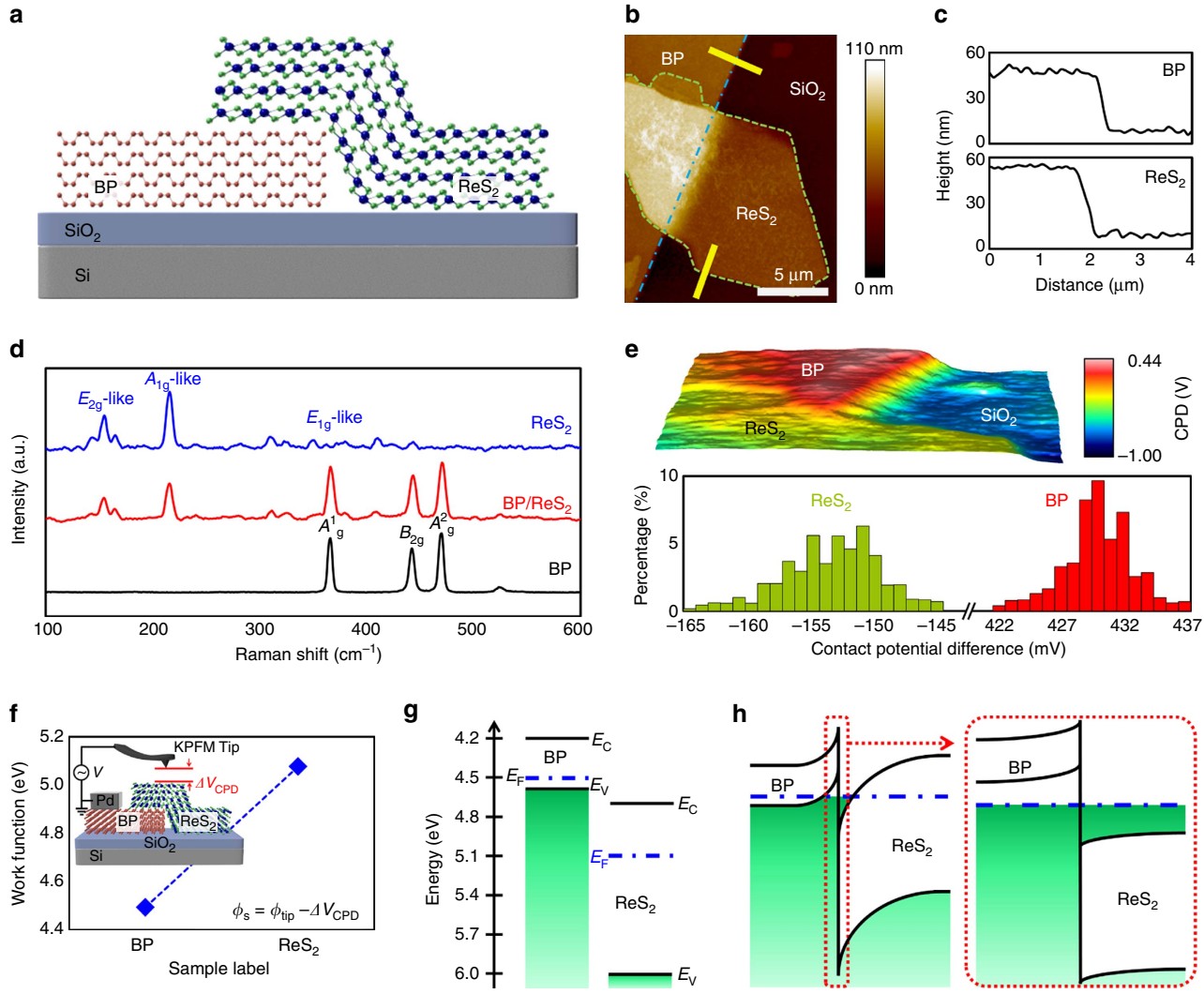

**Figure 1 | BP/ReS₂ heterostructure.** (**a**) Schematic illustration of the BP/ReS₂ heterostructure on SiO₂/Si substrate. (**b**) AFM (atomic force microscope) image of the BP/ReS₂ heterostructure sample. (**c**) Thicknesses of the BP (top) and the ReS₂ flakes (bottom) corresponding to the yellow lines marked in **b**. (**d**) Raman spectra of the ReS₂, BP/ReS₂ overlapped and BP regions. (**e**) Three-dimensional KPFM mapping image of the BP/ReS₂ heterostructure (top) and histogram distributions of $\Delta V_{CPD}$ extracted from the KPFM mapping image (bottom). (**f**) Work function values of BP and ReS₂ films. The inset shows schematic illustration of the KPFM measurement system. (**g,h**) Energy band alignments of BP and ReS₂ heterojunction at equilibrium (**g**) before and (**h**) after contact. $E_C$, $E_F$ and $E_V$ are the lowest energy level of the conduction band, the Fermi level and the highest energy level of the valence band of the semiconductors, respectively.

Furthermore, owing to the large work function difference (0.6 eV) between BP and ReS₂, hole and electron carriers accumulate near the heterojunction interface in BP and ReS₂, respectively (Fig. 1h). Therefore, a highly doped n⁺/p⁺ heterojunction can easily be implemented by forming a broken-gap band alignment without using a separate doping process, such as electrostatic doping by gate bias or chemical doping, which is generally required in a type-II heterojunction to realize a NDR device[2–4,6,7,33–36].

**BP/ReS₂ heterojunction-based NDR device.** After fabricating the NDR device based on the BP/ReS₂ heterojunction, as shown in Fig. 2a, we performed electrical measurements in the NDR device at room temperature. Figure 2b shows the current–voltage (I–V) characteristic of the NDR device on a linear scale. Here, the NDR behaviour was observed between 0.4 V and 0.9 V with a PVCR of 4.2, which is the highest value in previously reported NDR devices based on 2D materials[5–7,16–19]. We also note that similar electrical characteristics were observed in three different

BP/ReS₂ NDR devices with PVCR values between 3.8 and 4.1 (Inset of Fig. 2 and Supplementary Fig. 3). In addition, to understand the operation mechanism of the BP/ReS₂ NDR device, we theoretically investigated the current characteristic by considering tunnelling and diffusion currents using a theoretical model that we developed. The equations related to the current transport mechanisms can be found in Supplementary Fig. 4 and the parameters used in the analytic model are tabulated in Supplementary Table 1. The experimentally measured and theoretically calculated I–V curves are shown in Fig. 2c. Under a negative voltage and a positive voltage between 0 and 0.7 V, the tunnelling current seems to dominate the diffusion current, whereas the diffusion current primarily contributes to the operation of the NDR device when a higher voltage is applied (above 0.7 V). This is graphically explained in Fig. 2d, which shows the band alignments of the BP/ReS₂ heterojunction under various bias conditions. When a negative voltage is applied (V < 0 V), electron carriers are able to tunnel from the filled valence band states in BP to the empty conduction band states in

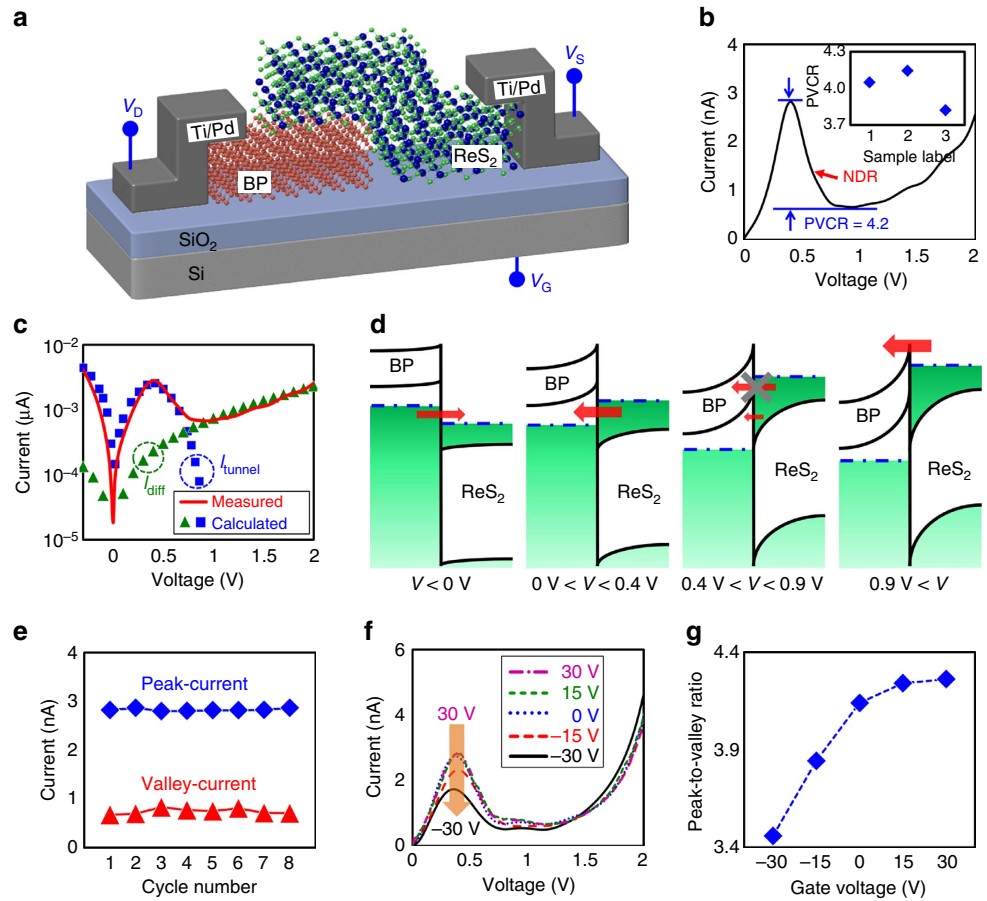

**Figure 2 | Electrical characteristics of BP/ReS$_2$ heterojunction-based NDR device at room temperature.** (**a**) An illustration of the BP/ReS$_2$ heterojunction NDR device. (**b**) Current–voltage (I–V) characteristic of the BP/ReS$_2$ NDR device on a linear scale. The inset shows the PVCR values for the three different BP/ReS$_2$ NDR devices. (**c**) Experimentally measured and theoretically calculated I–V curves of the BP/ReS$_2$ NDR device on a log scale. (**d**) Energy band alignment of the BP/ReS$_2$ heterojunction under various bias conditions. Width of the red arrow presents the magnitude of the current. (**e**) Extracted peak- and valley-current values of the BP/ReS$_2$ NDR device in eight consecutive I–V sweeps. (**f**) Drain current–drain voltage (I$_D$–V$_D$) curves under various gate biases from 30 V to −30 V. (**g**) PVCR values of the BP/ReS$_2$ NDR device as a function of gate voltage.

ReS$_2$, consequently increasing the current. Similarly, when a small positive voltage is applied (0 V < V < 0.4 V), the current increases because the electron carriers in the conduction band states of ReS$_2$ are tunnelled into the empty valence band states of BP. This current of the NDR device continuously increases until the Fermi level of ReS$_2$ aligns with the highest valence band energy of BP, where the filled states in the ReS$_2$ are maximally overlapped with unoccupied states of the BP, inducing a maximum tunnelling current (peak current). Further increases in voltage (0.4 V < V < 0.9 V) lead to decreases in the current because the degree of overlap between the filled and empty states is reduced due to the bandgap region. Therefore, the tunnelling current decreases with increasing voltage, and the NDR behaviour is obtained as shown in Fig. 2b,c. When a high voltage is applied (V > 0.9 V), the tunnelling current no longer affects the operation of the NDR device, and the electron carriers are able to diffuse from ReS$_2$ to BP by shrinking the potential hill in the BP/ReS$_2$ heterojunction, consequently again increasing the current of the BP/ReS$_2$ NDR device. Here, the lowest current value that is observed beyond a peak current is called a valley current. We then extracted the peak- and valley-current values of the NDR device for eight consecutive I–V sweeps, where stable peak- and valley-current values were observed, as shown in Fig. 2e. Figure 2f shows the drain current–drain voltage (I$_D$–V$_D$) curves under various gate bias conditions, which also confirms that the peak

current decreases as the gate voltage decreases. When the gate voltage varied from 30 V to −30 V, the Fermi level of BP down-shifted due to the accumulation of hole carriers, which thereby increased the degree of the energy band bending in the BP region (Supplementary Fig. 5). The Fermi level of the stacked ReS$_2$ on the BP is predicted to be barely modulated by an applied gate bias due to the thick BP (strong electrostatic screening effect). The down-shifted energy band in the BP region would form a potential well at the heterojunction interface, where much higher potential barrier height was obtained[45]. This leads to a decrease in the peak current of the BP/ReS$_2$ NDR device with decreasing gate voltage because strongly confined electron carriers in the potential well are difficult to escape from the potential well. The reduction of peak current in BP/ReS$_2$ NDR devices with decreasing gate voltage could also be estimated using the I$_D$–V$_D$ curves calculated by the analytic model (Supplementary Fig. 5). Thus, the PVCR of the BP/ReS$_2$ NDR device was modulated between 4.26 and 3.46 A/A by applying different gate voltages, as shown in Fig. 2g.

Furthermore, to analyse the temperature dependency of the carrier transport in the BP/ReS$_2$ NDR device, we performed I–V measurements at various temperatures between 180 and 300 K. As shown in Fig. 3a, the peak current (I$_{peak}$) increased, whereas the valley current (I$_{valley}$) decreased, with reducing temperature, consequently improving PVCR value from 4.02 to 6.78 A/A

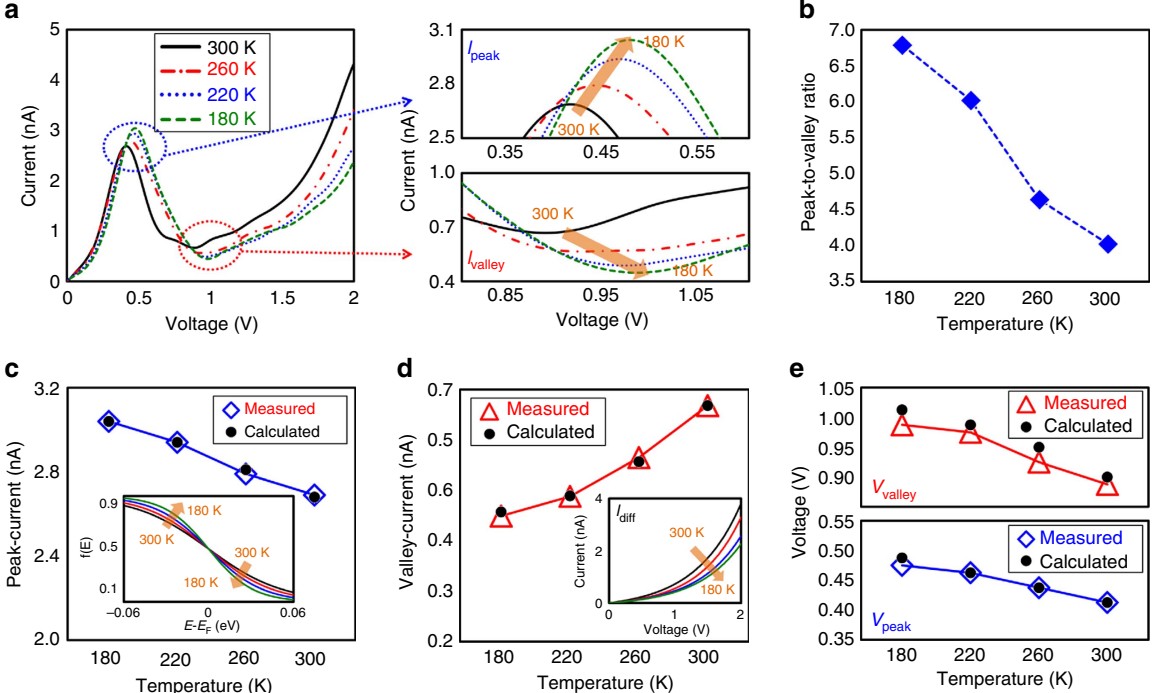

**Figure 3 | Temperature-dependent electrical characteristics of BP/ReS$_2$ NDR device.** (**a**) I–V curves of the BP/ReS$_2$ NDR device at various temperatures between 180 K and 300 K. (**b**) PVCR values of the BP/ReS$_2$ NDR device as a function of temperature. (**c**-**e**) Peak-current (**c**), valley-current (**d**), valley- and peak-voltage values of the BP/ReS$_2$ NDR device as a function of temperature (**e**), which were extracted from the experimentally measured and the theoretically calculated I–V characteristic curves. The inset in **c** shows the probability of states being occupied (f(E)) as a function of given energy E relative to E$_F$(E − E$_F$). The inset in **d** shows the theoretically calculated diffusion current of the BP/ReS$_2$ NDR device at various temperatures.

(Fig. 3b). In addition, the peak-voltage ($V_{peak}$) and valley-voltage ($V_{valley}$) values shifted positively as the measurement temperature was reduced. To quantitatively analyse the temperature-dependent electrical characteristics of the BP/ReS$_2$ NDR device, we exploited the proposed analytic NDR device model. The calculated I–V characteristic curves at different temperatures are presented in Supplementary Fig. 6, where the I–V curves estimated by the analytic model were well fitted with the measured I–V data. Figure 3c shows the $I_{peak}$ data as a function of temperature, which were extracted from the experimentally measured and the theoretically calculated I–V characteristics. Because a large portion of $I_{peak}$ is mainly occupied by $I_{tunnel}$, as shown in Fig. 2c, the $I_{peak}$ seems to be associated with the density of states in the conduction band of ReS$_2$ and the valence band of BP, where the density of occupied or empty states is determined by the Fermi–Dirac function. Thus, we focused on an analysis on the temperature dependency of the Fermi–Dirac distribution. As the temperature decreases, the Fermi–Dirac distribution near the Fermi level of BP and ReS$_2$ becomes sharp, thereby increasing the probability of states being occupied (f(E)) in the conduction band of ReS$_2$, as shown in the inset of Fig. 3c, where f(E) at energy E of $E_F − 0.03$ eV were 0.76 and 0.87 at 300 K and 180 K, respectively. Meanwhile, the f(E) in the valence band of BP decreases (thereby, an increase in probability of states being empty) with reducing temperature, where f(E) at energy $E = E_F + 0.03$ eV were 0.24 and 0.13 at 300 and 180 K, respectively. This subsequently increases $I_{tunnel}$ because of the increased occupied states in conduction band of ReS$_2$ and the decreased empty states in valence band of BP, eventually resulting in a slight increase of $I_{peak}$ (2.7 nA at 300 K and 3.0 nA at 180 K in Fig. 3c). In contrast, because the dominant current of $I_{valley}$ is $I_{diff}$, which is dependent on temperature (see the inset of Fig. 3d), $I_{valley}$ is predicted to reduce with decreasing temperature (0.67 nA at 300 K and

0.45 nA at 180 K, in Fig. 3d). Overall, in the BP/ReS$_2$ NDR device, the temperature dependencies of $I_{peak}$ and $I_{valley}$ were differently presented due to the increase in $I_{tunnel}$ and the decrease in $I_{diff}$ as the measurement temperature decreased. Meanwhile, we also considered parasitic series resistance ($R_s$) in the analytic model to accurately analyse the device operation. $R_s$ is mainly associated with the contact resistance between the metal electrode and the semiconductor[46]. Here, the reduction in n-type carrier concentration due to decreasing temperature leads to an increase in the depletion width at the metal/semiconductor (MS) junction and thereby a suppression of the e-field-dependent barrier height lowering effect, eventually increasing the contact resistance at the MS junction (Supplementary Fig. 7)[47]. Thus, as shown in Fig. 3e, positively shifted $V_{peak}$ and $V_{valley}$ were observed as measurement temperature decreased because a higher voltage was required to operate the NDR device due to the increased $R_S$ at reduced temperature.

**Ternary inverter with three logical states.** Finally, we fabricated a ternary inverter, which is a basic building block in MVL applications, as schematically shown in Fig. 4a. This ternary inverter was formed by integrating the BP/ReS$_2$ heterojunction NDR device as a driver with the built-in BP p-channel TFT as a load resistor, where the total resistance in the BP TFT could be controlled by an applied gate voltage (Supplementary Fig. 8). Figure 4b,c show the equivalent circuit configuration and an optical image of the ternary inverter, respectively. The supply ($V_{DD}$) and input voltages ($V_{IN}$) were applied to the source electrode on the BP and the back gate. The metal electrode on the ReS$_2$ (source electrode in the BP/ReS$_2$ NDR device) was connected to the ground ($V_{SS}$), and then we measured the output voltage ($V_{OUT}$) on the middle shared electrode (drain electrode in

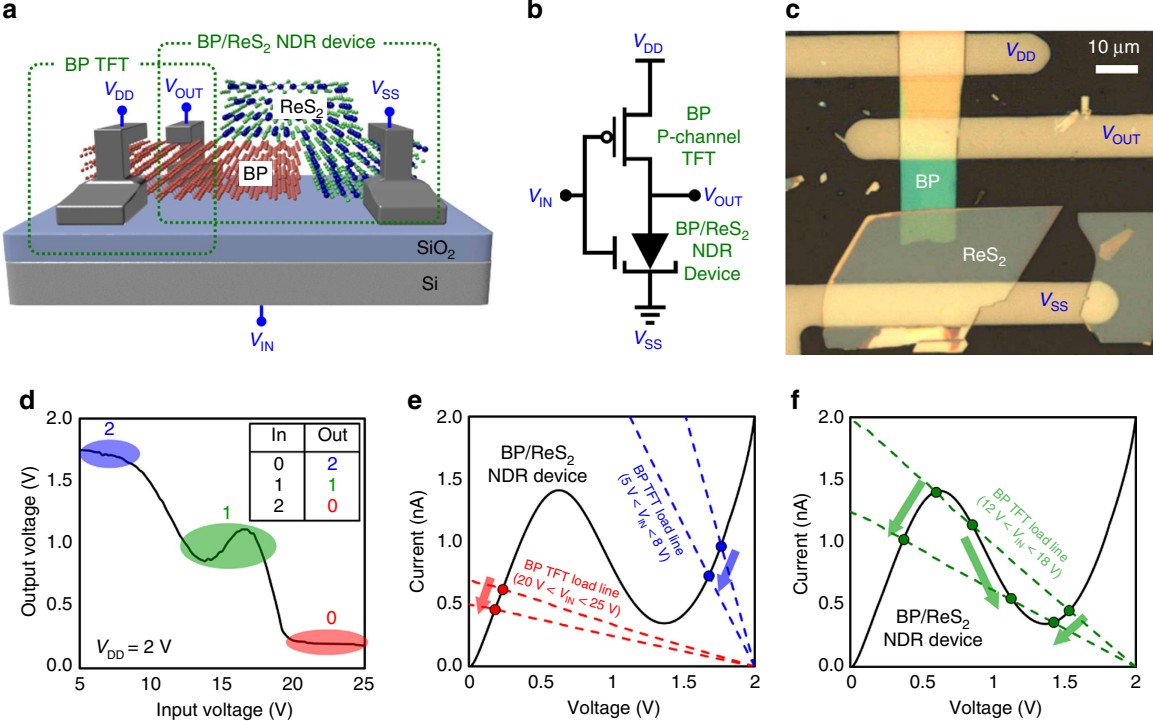

**Figure 4 | Ternary inverter with three logical states.** (**a**) Schematic illustration of the ternary inverter. (**b**) Equivalent circuit configuration of the ternary inverter. (**c**) Optical image of the ternary inverter. (**d**) $V_{IN}$ versus $V_{OUT}$ characteristic of the ternary inverter. The inset shows an input–output table of the ternary inverter. (**e,f**) Load-line analysis of the ternary inverter circuit under three bias conditions: (**e**) $5 \, V < V_{IN} < 8 \, V$, $20 \, V < V_{IN} < 25 \, V$ and (**f**) $12 \, V < V_{IN} < 18 \, V$. The $I$–$V$ characteristics of the BP/ReS$_2$ NDR device (driver) and the BP TFT (load resistor) are represented by solid and dashed lines, respectively.

the BP TFT and in the BP/ReS$_2$ NDR device). The $V_{IN}$ versus $V_{OUT}$ characteristic of the ternary inverter is shown in Fig. 4d, where $V_{DD}$ was 2 V. When $V_{IN}$ varied from 5 V to 25 V, $V_{OUT}$ showed three distinct states: (i) $V_{OUT} > 1.7 \, V$ (state '2') for $5 \, V < V_{IN} < 8 \, V$, (ii) $0.85 \, V < V_{OUT} < 1.12 \, V$ (state '1') for $12 \, V < V_{IN} < 18 \, V$ and (iii) $V_{OUT} < 0.24 \, V$ (state '0') for $20 \, V < V_{IN} < 25 \, V$. To explain the operation of this ternary inverter, we performed a load-line circuit analysis, in which the intersections of the two characteristic curves indicate the operating points of this circuit. As shown in Fig. 4e, when a low $V_{IN}$ is applied ($5 \, V < V_{IN} < 8 \, V$), the load resistor (BP TFT) provides a low-resistance path between the source ($V_{DD}$) and drain (output) nodes of the BP TFT because the applied $V_{IN}$ is higher than the threshold voltage ($V_{TH}$) of the BP TFT (Supplementary Fig. 8). Thus, high voltage values (logic state '2'), which were close to $V_{DD}$, were measured at the output terminal (blue circles in Fig. 4e). In contrast, when a high $V_{IN}$ was applied ($20 \, V < V_{IN} < 25 \, V$), the BP TFT was turned off ($V_{IN} < V_{TH}$), which creates a low-resistance path between the output terminal and the ground. This consequently presented low voltage values (logic state '0') at the output terminal (red circles in Fig. 4e). When a moderate $V_{IN}$ was applied ($12 \, V < V_{IN} < 18 \, V$), the operating points were located at the NDR region in the $I$–$V$ curve of the BP/ReS$_2$ NDR device, as shown in Fig. 4f. This resulted in intermediate output values (logic state '1') with small fluctuations due to an imbalance of the operating points, where the three intersections were. Overall, by integrating the BP/ReS$_2$ NDR device with the built-in BP TFT, the ternary inverter was simply demonstrated as an MVL application.

## Discussion

We demonstrated a NDR device based on a BP/ReS$_2$ heterojunction with high PVCR values of 4.2 and 6.8 at room temperature

and 180 K, respectively. This NDR characteristic can be easily achieved by forming a broken-gap (type-III) band alignment without a separate process, such as electrostatic doping by gate bias and a chemical doping process, which is generally required in type-II heterojunction to realize an NDR device. The broken-gap band alignment of the BP/ReS$_2$ heterojunction was confirmed through KPFM measurements, where the band gaps of the BP and ReS$_2$ did not overlap at all (type-III). Also, the carrier transport mechanisms of the BP/ReS$_2$ NDR device were investigated in detail by analysing the tunnelling and diffusion currents at various temperatures between 180 and 300 K by using the proposed analytic NDR device model. Specifically, we confirmed that $I_{peak}$ increased while $I_{valley}$ decreased as the measurement temperature was reduced, consequently providing a PVCR value that improved from 4.02 to 6.8. Finally, we demonstrated a ternary inverter as an MVL application, which was fabricated by integrating a BP/ReS$_2$ heterojunction NDR device with a built-in BP TFT. In the $V_{IN}$ versus $V_{OUT}$ characteristic of the ternary inverter, when $V_{IN}$ varied from 5 to 25 V, $V_{OUT}$ showed three distinct values (states '2', '1' and '0'). This study of a 2D material heterojunction is a step forward toward future multi-valued logic device research.

## Methods

**Fabrication of the BP/ReS$_2$ heterojunction-based NDR devices.** A BP flake was exfoliated onto a 90 nm thick SiO$_2$/Si substrate by adhesive tape (224SPV, Nitto). Then, a ReS$_2$ flake was transferred onto the BP flake by using a mechanical transfer process method. Finally, the source and drain electrode regions were patterned by optical lithography, and Ti/Pd (10/30 nm) layers were deposited on an electron-beam evaporating system, followed by a lift-off process.

**Fabrication of the ternary inverter.** By using a mechanical transfer method, a ReS$_2$ flake was stacked onto the BP flake, which was exfoliated onto a 90 nm thick SiO$_2$/Si substrate. The metal electrode regions were defined using a conventional photolithography process. Finally, Ti/Pd (10/30 nm) layers were deposited by

e-beam evaporation to form the contacts for BP and $ReS_2$, followed by a lift-off process in acetone. The BP/$ReS_2$ NDR and the BP TFT devices were designed to function as a driver and a load resistor for a ternary inverter, respectively. The voltage of $V_{DD}$ was applied to the source electrode of the BP TFT, and the source electrode of the BP/$ReS_2$ NDR device was connected to the ground ($V_{SS}$). The common back gate of the BP TFT and BP/$ReS_2$ NDR devices served as the input voltage ($V_{IN}$) electrode. The output voltage ($V_{OUT}$) was measured at the drain electrode of the BP/$ReS_2$ NDR device.

**Characterization of the BP/$ReS_2$ heterojunctions.** Raman studies were conducted using a WITec micro-Raman spectrometer system with a frequency-doubled Nd-YAG laser beam (532 nm laser excitation). The atomic force microscope analysis was carried out in an XE 100 (Park Systems Corp.) system. The electrical transport measurements were conducted at room temperature under ambient conditions in a probe station with a Keysight B2912A. The temperature-dependent electrical characteristics were measured in a vacuum chamber (below $10^{-4}$ Torr) using a Keithley 4200 Semiconductor Parameter Analyzer. The KPFM measurement was performed using NTEGRA Spectra (NT-MDT).

**Theoretic model of carrier transport in BP/$ReS_2$ heterojunctions.** The tunnelling current ($I_{tunnel}$) and diffusion current ($I_{diff}$) were considered to understand the operating mechanism of the BP/$ReS_2$ NDR device. The $I_{tunnel}$ can be obtained from

$$I_{tunnel} = \frac{2\pi\alpha q}{h} \int_{E_{C\_Re}}^{E_{V\_BP}} DOS_{BP}(E) \times DOS_{Re}(E) \times [f_{BP}(E) - f_{Re}(E - q(V - IR_s))]dE$$

(1)

where $\alpha$ is the screening factor, $q$ is the elementary charge, $h$ is the Planck constant, $E_{V\_BP}$ is the highest valence band energy in BP, $E_{C\_Re}$ is the lowest conduction band energy in $ReS_2$. $DOS_{BP}(E)$, $DOS_{Re}(E)$, $f_{BP}(E)$ and $f_{Re}(E)$ mean the density of states and Fermi–Dirac distribution functions of BP and $ReS_2$, respectively.

The $I_{diff}$ is obtained from

$$I_{diff} = qI_0 \left[ \exp\left( \frac{V - IR_s}{\eta_{id}k_B T} \right) - 1 \right]$$

(2)

where $I_0$ is the saturation current, $V$ is the applied voltage, $I$ is the junction current, $R_s$ is the series resistance, $\eta_{id}$ is the ideality factor, $k_B$ is the Boltzmann constant and $T$ is the temperature.

**Data availability.** The data that support the findings of this study are available from the corresponding author upon request.

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

## Acknowledgements

This work was supported by the National Research Foundation of Korea (NRF) funded by the Korea government (MSIP) (No. 2015R1A2A2A01002965).

## Author contributions

J.S. and J.-H.P. conceived and designed the experiments. J.S., S.O., D.-H.K., S.-H.J., M.H.A., W.-Y.C. and K.H. contributed to the experimental design and device fabrication. J.S. analysed the data and performed the theoretical calculations. J.J. and S.L. conducted the Raman study and data analysis. M.K. and Y.J.S. performed the KPFM measurement. J.-H.P. supervised the research. All the authors discussed the results and commented on the manuscript.

## Additional information

**Competing financial interests:** The authors declare no competing financial interests.

