## [Peer Review File · Nature Communications]

Reviewers' comments:

Reviewer #1 (Remarks to the Author):

The authors in this manuscript reported a new heterojunction device with negative differential resistance (NDR), namely black phosphorous-ReS₂ van der Waals heterostructure. Although 2D-2D heterostructures with similar NDR behavior were reported before, as cited in this manuscript, the device performance here in terms of NDR behavior is best than anything in the literature. The amount of work reported here, such as the KPFM characterization, Raman spectroscopy, low temperature measurement, is comprehensive. The demonstration of the ternary inverter using the NDR diode based logic circuit highlights the potential of the BP-ReS₂ type III heterojunction. Overall, I think that this manuscript merits publication in Nature Communication. Addressing the following minor issues however, will increase the impact and the readership of this paper.

1) in Figure 1g and the associated text in the manuscript, there seems to be no discussion about the origin for the values of $(E_f - E_v)$ in BP and $(E_c - E_f)$ in ReS₂. Are those values measured, or cited from literature? The authors should clarify this point.

2) the high peak-to-valley current ratio (PVCR) is the highlight of this work, and it is very good that this manuscript reports 3 or more devices in total having high PVCR. Can the authors show some statistics of this important demonstration in the main Figure instead of the supplementary information?

3) the scale bar in Figure 4c is missing.

4) the literature on the 2D-2D semiconductor heterostructure is not well cited in the manuscript. For example, PNAS April 29, 2014 vol. 111 no. 17 pp 6198-6202; Nature Nanotechnology 9, 676-681 (2014); Nature Nanotechnology 9, 682-686 (2014); Nano Lett., 2014, 14 (8), pp 4785-4791 etc.

Reviewer #2 (Remarks to the Author):

1. The author strongly states that the broken-band alignment of BP/ReS₂ heterojunction is confirmed through the KPFM measurement. However, KPFM data shown in the manuscript only gives information on work function of each material, not on CBM and VBM. The author seems to complete the band structure of BP/ReS₂ heterojunction (figure.1g) by utilizing the theoretically calculated data of reference 36 and 37, but I think those data are not fully matched to the 40nm BP and 50nm ReS₂ (flakes utilized at the device). As the author has emphasized that the broken-band alignment is key factor to fabricating NDR easily, KPFM data seems not enough to prove that point

2. Figure. 1f seems too complicated, so it makes hard to figure out the data and explanation on the configuration of KPFM measurement.

3. 3. In the manuscript, experimental method on ternary inverter fabrication is not well-described. I suggest to add more explanation on the fabrication process and related experimental details

General Review: In this paper, the author has fabricated a BP/ReS₂ heterojunction based NDR device showing high PVCR value and also demonstrated ternary inverter as a multi-valued logic application. As the author has exhibited high-performance NDR device using 2D material and also conducted systematic analysis on its charge transport mechanism, I think this manuscript is valuable enough to be published just with minor tuning on the points mentioned above.

Reviewer #3 (Remarks to the Author):

The manuscript presents an interesting study of negative differential resistance in black phosphorus/ReS₂ heterojunction, and a demonstration of a multi-value logic (MVL) circuitry using one of the semiconductors as a transistor loaded by the tunnel diode. The material system chosen for this study is indeed new (Ref 5 reported a black phosphorus/SnSe₂ tunnel diode with lower peak-to-valley-ratio); and the degree of NDR is improved from this earlier report. The MVL is old in traditional semiconductors, and is similar to the one reported in Ref 7. The results are indicative of heterojunction formation and interband tunneling; the models reported are able to explain the observed experimental work and are similar to prior (cited) work, and are not newly developed for this work.

The work is of high interest, and shows the richness of 2D layered material system choices. It improves on prior work (which is very important), the experiments and models presented here do not represent a new finding.

Some of the claims in the manuscript could be misleading and should be corrected in future versions:

1) In the introduction (page 4), it is claimed that "most NDR devices have shown very low PVCR .. at low temperatures... heterojunctions interfaces suffered significantly from defects...". This is incorrect. SiGe and III-V heterojunctions show robust, very high PVCRs at room temperature, much higher and more reproducible than what is reported in this manuscript and most 2D layered materials. Many of them are used in commercial devices, for example broken-gap III-V backward diodes are used for millimeter wave imaging applications. MVL logic has been demonstrated with 3D tunnel junctions, it just has not been very useful.

These sorts of claims should be carefully considered because sometimes the 2D materials community do not do proper research of existing 3D semiconductor capabilities and make erroneous claims.

2) In page 4, the claim of " ... electrostatic doping ... " is incorrect. Any useful electronic device will need to be doped to be controlled; it does not help in adjusting threshold voltages etc if the doping is left to nature. This claim is repeated in page 6. This is a fundamental underestimation of the importance of doping in device physics.

3) Because the thickness of the BP layer is so high, how is the gating affecting the conductance? If it is changing the conductivity of the entire channel, it should reach in and also affect the band alignments, etc. This is not included in the analysis or model.

4) Page 6: PVCRs exceeding the reported values here by orders of magnitude have been observed in 3D semiconductor tunnel junctions.

5) The modeling in the supplemental material must provide the parameters that were used, they are missing.

Point by Point Response to the Reviewers

Reviewer: 1 Comments to the Author

The authors in this manuscript reported a new heterojunction device with negative differential resistance (NDR), namely black phosphorous-ReS₂ van der Waals heterostructure. Although 2D-2D heterostructures with similar NDR behavior were reported before, as cited in this manuscript, the device performance here in terms of NDR behavior is best than anything in the literature. The amount of work reported here, such as the KPFM characterization, Raman spectroscopy, low temperature measurement, is comprehensive. The demonstration of the ternary inverter using the NDR diode based logic circuit highlights the potential of the BP-ReS₂ type III heterojunction. Overall, I think that this manuscript merits publication in Nature Communication. Addressing the following minor issues however, will increase the impact and the readership of this paper.

We would like to thank you for reviewing our paper; we appreciate your insightful comments on our research. We have revised the manuscript according to your suggestions and believe that the revisions have improved the paper.

Please find below our responses (in blue) to each specific comment (in black) provided by the reviewer. In addition, revisions to the original article are highlighted in red.

(1) In Figure 1g and the associated text in the manuscript, there seems to be no discussion about the origin for the values of (E_f-E_v) in BP and (E_c-E_f) in ReS₂. Are those values measured, or cited from literature? The authors should clarify this point.

ANS: We obtained the conduction band minimum (CBM), the valence band maximum (VBM), and the band gap (E_g) values of the BP and ReS₂ from previously reported literature⁴⁰⁻⁴².

To clarify this point, we modified the following sentences and added the references in the article.

“Based on the obtained KPFM results and the previously reported band properties (conduction band minimum (CBM), valence band maximum (VBM), and band gap (E_g)) of BP and ReS₂⁴⁰⁻⁴², we graphically described the predicted energy band alignment of the BP and ReS₂ heterojunction at

equilibrium before contact (Fig. 1g) and after contact (Fig. 1h). Here, the CBM, VBM, and E_g values of the BP (ReS_2) that were calculated using a first-principles density of states in the literature were 4.2 eV (4.68 eV), 4.59 eV (6.05 eV), and 0.39 eV (1.37 eV), respectively.”

[40] Perello, D. J., Chae, S. H., Song, S. & Lee, Y. H. High-performance n-type black phosphorus transistors with type control via thickness and contact-metal engineering. *Nat. Commun.* **6**, 7809 (2015).

[41] Liu, X. *et al.* Black Phosphorus Based Field Effect Transistors with Simultaneously Achieved Near Ideal Subthreshold Swing and High Hole Mobility at Room Temperature. *Sci. Rep.* **6**, 24920 (2016).

We also modified Figure 1g as below.

(2) The high peak-to-valley current ratio (PVCR) is the highlight of this work, and it is very good that this manuscript reports 3 or more devices in total having high PVCR. Can the authors show some statistics of this important demonstration in the main Figure instead of the supplementary information?

ANS: As the reviewer suggested, we added the PVCR values for the three different BP/ ReS_2 NDR devices in Figure 2b.

We also modified the following sentences in the article.

In the figure caption:

(a) An illustration of the BP/ReS₂ heterojunction NDR device. **(b)** Current–voltage (*I*-*V*) characteristic of the BP/ReS₂ NDR device on a linear scale. The inset shows the PVCR values for the three different BP/ReS₂ NDR devices.

In the article:

“We also note that similar electrical characteristics were observed in three different BP/ReS₂ NDR devices with PVCR values between 3.8 and 4.1 (Inset of Fig. 2 and Supporting Information Fig. S3).”

(3) The scale bar in Figure 4c is missing.

ANS: We added the scale bar in Figure 4c, as shown below.

(4) The literature on the 2D-2D semiconductor heterostructure is not well cited in the manuscript. For example, PNAS April 29, 2014 vol. 111 no. 17 pp 6198-6202; Nature Nanotechnology 9, 676-681 (2014); Nature Nanotechnology 9, 682-686 (2014); Nano Lett., 2014, 14 (8), pp 4785-4791 etc.

ANS: As the reviewer suggested, we added the references as below:

“It is also possible to design various heterojunctions by stacking different 2D materials with different bandgaps and electron affinities, where band structure alignment can be classified into three types: type-I (straddling-gap)³⁵, type-II (staggered-gap)^{2-4,6,7,31-34}, and type-III (broken-gap)^{5,35}.”

[31] Lee, C.-H. *et al.* Atomically thin p–n junctions with van der Waals heterointerfaces. *Nat. Nanotechnol.* **9**, 676-681 (2014).

[32] Fang, H. *et al.* Strong interlayer coupling in van der Waals heterostructures built from single-layer chalcogenides. *PNAS* **111**, 6198-6202 (2014).

[33] Furchi, M. M., Pospischil, A., Libisch, F., Burgdörfer, J. & Mueller, T., Photovoltaic Effect in an Electrically Tunable van der Waals Heterojunction. *Nano Lett.* **14**, 4785-4791 (2014).

[34] Hong, X. *et al.* Ultrafast charge transfer in atomically thin MoS₂/WS₂ heterostructures. *Nat. Nanotechnol.* **9**, 682-686 (2014).

Reviewer: 2 Comments to the Author

In this paper, the author has fabricated a BP/ReS₂ heterojunction based NDR device showing high PVCR value and also demonstrated ternary inverter as a multi-valued logic application. As the author has exhibited high-performance NDR device using 2D material and also conducted systematic analysis on its charge transport mechanism, I think this manuscript is valuable enough to be published just with minor tuning on the points mentioned above.

We would like to thank you for reviewing our paper; we appreciate your insightful comments on our research. We have revised the manuscript according to your suggestions and believe that the revisions have improved the paper.

Please find below our responses (in blue) to each specific comment (in black) provided by the reviewer. In addition, revisions to the original article are highlighted in red.

(1) The author strongly states that the broken-band alignment of BP/ReS₂ heterojunction is confirmed through the KPFM measurement. However, KPFM data shown in the manuscript only gives information on work function of each material, not on CBM and VBM. The author seems to complete the band structure of BP/ReS₂ heterojunction (figure.1g) by utilizing the theoretically calculated data of reference 36 and 37, but I think those data are not fully matched to the 40nm BP and 50nm ReS₂ (flakes utilized at the device). As the author has emphasized that the broken-band alignment is key factor to fabricating NDR easily, KPFM data seems not enough to prove that point.

ANS: As the reviewer pointed out, we obtained only work function values of the materials through the KPFM measurement. We then referred to the conduction band minimum (CBM), the valence band maximum (VBM), and the band gap (E_g) values of the bulk BP and ReS₂ from previously reported literature⁴⁰⁻⁴² (Here, we added new references that reported the CBM, VBM, and E_g values of bulk BP).

To clarify this point, we modified the following sentences and added the references in the article.

“Based on the obtained KPFM results and the previously reported band properties (conduction band minimum (CBM), valence band maximum (VBM), and band gap (E_g)) of BP and ReS_2 ⁴⁰⁻⁴², we graphically described the predicted energy band alignment of the BP and ReS_2 heterojunction at equilibrium before contact (Fig. 1g) and after contact (Fig. 1h). Here, the CBM, VBM, and E_g values of the BP (ReS_2) that were calculated using a first-principles density of states in the literature were 4.2 eV (4.68 eV), 4.59 eV (6.05 eV), and 0.39 eV (1.37 eV), respectively.”

[40] Perello, D. J., Chae, S. H., Song, S. & Lee, Y. H. High-performance n-type black phosphorus transistors with type control via thickness and contact-metal engineering. *Nat. Commun.* **6**, 7809 (2015).

[41] Liu, X. *et al.* Black Phosphorus Based Field Effect Transistors with Simultaneously Achieved Near Ideal Subthreshold Swing and High Hole Mobility at Room Temperature. *Sci. Rep.* **6**, 24920 (2016).

We also modified Figure 1g as below.

(2) Figure. 1f seems too complicated, so it makes hard to figure out the data and explanation on the configuration of KPFM measurement.

ANS: We modified Figure 1f to facilitate a better understanding of the KPFM analysis results for the BP/ ReS_2 heterojunction.

(3) In the manuscript, experimental method on ternary inverter fabrication is not well-described. I suggest to add more explanation on the fabrication process and related experimental details.

ANS: As the reviewer suggested, we explained the fabrication process flow of the ternary inverter in the “experimental methods” section as below:

“Fabrication of the ternary inverter. By using a mechanical transfer method, a ReS₂ flake was stacked onto the BP flake, which was exfoliated onto a 90 nm thick SiO₂/Si substrate. The metal electrode regions were defined using a conventional photolithography process. Finally, Ti/Pd (10/30 nm) layers were deposited by e-beam evaporation to form the contacts for BP and ReS₂, followed by a lift-off process in acetone. The BP/ReS₂ NDR and the BP TFT devices were designed to function as a driver and a load resistor for a ternary inverter, respectively. The voltage of V_{DD} was applied to the source electrode of the BP TFT, and the source electrode of the BP/ReS₂ NDR device was connected to the ground (V_{SS}). The common back gate of the BP TFT and BP/ReS₂ NDR devices served as the input voltage (V_{IN}) electrode. The output voltage (V_{OUT}) was measured at the drain electrode of the BP/ReS₂ NDR device.”

Reviewer: 3 Comments to the Author

We would like to thank you for reviewing our paper; we appreciate your insightful comments on our research. We have revised the manuscript according to your suggestions and believe that the revisions have improved the paper.

Please find below our responses (in blue) to each specific comment (in black) provided by the reviewer. In addition, revisions to the original article are highlighted in red.

The manuscript presents an interesting study of negative differential resistance in black phosphorus/ReS₂ heterojunction, and a demonstration of a multi-value logic (MVL) circuitry using one of the semiconductors as a transistor loaded by the tunnel diode. The material system chosen for this study is indeed new (Ref 5 reported a black phosphorus/SnSe₂ tunnel diode with lower peak-to-valley-ratio); and the degree of NDR is improved from this earlier report. The MVL is old in traditional semiconductors, and is similar to the one reported in Ref 7. The results are indicative of heterojunction formation and interband tunneling; the models reported are able to explain the observed experimental work and are similar to prior (cited) work, and are not newly developed for this work.

ANS: To realize MVL circuits with traditional semiconductor (Si or III-V) devices, considerable number of devices were required (more than nine transistors or four RTDs were needed for ternary logic circuits)^{R.1-3}. Therefore, this kind of MVL circuit consisting of conventional CMOS devices or RTDs, from a practical standpoint, suffered from high power dissipation due to the increased parasitic capacitance of the interconnects.

[R.1] T. Waho *et al. IEEE J. Solid-State Circuits* **33**, 268-274 (1998).

[R.2] K.-J. Gan *et al. Solid-State Electronics* **54**, 1637-1640 (2010).

[R.3] J. Nunez *et al. IEEE International Symposium on Circuits and Systems (ISCAS 2008)* 604-607 (2008).

In this work, we proposed a new approach to realize a ternary inverter for MVL, where a BP/ReS₂ heterojunction-based NDR device and a built-in BP p-channel TFT were integrated. This approach, which is based on NDR devices, is expected to fulfil low-power advantages of future MVL circuits by reducing the parasitic interconnect capacitance.

Although our results are similar to the previously reported work (Ref. 7), which was conducted recently (published on January 2016) as our experiment, our work proposed a totally different method to demonstrate MVL circuits, as shown in the table below.

	This work	Ref. 7
Materials	BP/ReS ₂	WSe ₂ /MoS ₂
Heterojunction type	Type III broken-gap	Type II staggered-gap
Usable flake thickness	BP bulk (30-60 nm)/ ReS ₂ bulk (30-60 nm)	2 nm/10 nm (NDR behavior was only observed in this condition)
PVCR	4.2 at RT, 6.9 at 180 K	< 1.6 at RT
Analytic model	Considered tunneling/diffusion mechanisms and the effect of parasitic series resistance	Considered only tunneling mechanism
Operation principle of ternary inverter	Sign-change in output-conductance ($\partial I_b / \partial V_b$)	Sign-change in transconductance ($\partial I_b / \partial V_G$)

Firstly, because our BP/ReS₂ heterojunction was formed by a broken-gap band alignment, this type-III junction-based NDR device is independent of the thickness of the flakes. However, in the case of a WSe₂/MoS₂ heterojunction device (Ref. 7), which was formed by a staggered-gap band alignment, NDR behavior was only observed under specific conditions (2 nm WSe₂/10 nm MoS₂).

Secondly, only the tunneling mechanism was considered in the analytic model of Ref. 7. However, to investigate the carrier transport mechanism of our NDR device, we developed a new analytic model

that considers both tunneling/diffusion currents and parasitic series resistance, which could help elucidate the peak-/valley-currents and voltages of the NDR device, respectively.

Lastly, the ternary inverters in our work and Ref. 7 operated differently. In our work, a sign-change in output-conductance ($\partial I_D/\partial V_D$) was used, and in Ref. 7, a sign-change in transconductance ($\partial I_D/\partial V_G$) was used. This is evidence that we proposed a totally different method to demonstrate MVL circuits.

Based on the reviewer's comments, we've added and modified the following sentences in the article.

"In addition, as an MVL application, we present a ternary inverter (having three states) that combines a BP/ReS₂ heterojunction NDR device and a BP p-channel thin film transistor (TFT). **This integration approach based on NDR devices is expected to fulfil low-power advantages of future MVL circuits by reducing the parasitic interconnect capacitance. In particular,** compared to a type-II heterojunction, a type-III heterojunction can easily implement a highly doped n⁺/p⁺ heterojunction without a separate process, such as electrostatic doping by gate bias and a chemical doping process."

"Furthermore, through temperature-dependent current–voltage (*I-V*) measurements and the proposed analytic NDR device model, **where tunneling/diffusion currents and parasitic series resistance were considered simultaneously,** we quantitatively study the temperature-dependent device operations."

The work is of high interest, and shows the richness of 2D layered material system choices. It improves on prior work (which is very important), the experiments and models presented here do not represent a new finding.

Some of the claims in the manuscript could be misleading and should be corrected in future versions:

ANS: The authors think that this work has three critical points (*i.e.*, new findings).

- i) **Doping-free fabrication process using a type-III heterojunction:** The BP/ReS₂ heterojunction was formed by a broken-gap band alignment. Subsequently, based on the broken-gap band alignment (type-III heterojunction), a highly doped n⁺/p⁺ heterojunction was easily implemented without using a separate doping process, such as electrostatic doping by gate

bias or chemical doping, which is generally required in a type-II heterojunction to realize an NDR device.

- ii) **Quantitative analysis through development of an analytic NDR device model:** The carrier transport mechanism of the BP/ReS₂ heterojunction NDR device was investigated in detail at various temperatures between 180 K and 300 K with the developed analytic NDR device model, simultaneously considering both tunneling/diffusion currents and parasitic series resistance.
- iii) **New implementation of a ternary inverter:** As an MVL application expected to satisfy future low-power demands by reducing the parasitic capacitance of the interconnects, a ternary inverter could be fabricated by integrating the BP/ReS₂ heterojunction NDR device as a driver with the built-in BP p-channel TFT as a load resistor, where the total resistance in the BP TFT was controlled by an applied gate voltage.

(1) In the introduction (page 4), it is claimed that "most NDR devices have shown very low PVCR at low temperatures... heterojunctions interfaces suffered significantly from defects...". This is incorrect. SiGe and III-V heterojunctions show robust, very high PVCRs at room temperature, much higher and more reproducible than what is reported in this manuscript and most 2D layered materials. Many of them are used in commercial devices, for example broken-gap III-V backward diodes are used for millimeter wave imaging applications. MVL logic has been demonstrated with 3D tunnel junctions, it just has not been very useful.

These sorts of claims should be carefully considered because sometimes the 2D materials community do not do proper research of existing 3D semiconductor capabilities and make erroneous claims.

ANS: We agree with the reviewer comment that NDR devices based on SiGe or III-V heterojunctions exhibited reproducible NDR behavior with high PVCR values (above 10) at room temperature. We corrected this in the article as below.

However, it is still true that the formation of various types of heterojunctions (type-I, II, and III) on SiGe and III-V materials is limited by threading dislocations, which are normally caused at the junction interface by lattice mismatch during film growth. Although the threading dislocation that increases the valley current of the NDR device can be reduced through superlattice and nanowire structures, it is hard to avoid that the fabrication process becomes more complex.

Whereas, various high-quality heterojunctions can be formed without the lattice mismatch issue by just stacking 2D materials due to the absence of dangling bonds on their surface (The authors

believe that this is the most important advantage of 2D semiconductor materials compared to conventional 3D materials).

In order to clarify this, we modified and added the following sentences in the article.

~~“However, at the current stage of research, most NDR devices have shown very low peak-to-valley current ratio (PVCR), which is an important parameter for evaluating NDR device performance, and the NDR property was normally observed at low temperatures^{6,8,18,20,21,23}. Furthermore, because most of the Esaki diodes and RTDs were fabricated in Si-Ge and III-V semiconductors^{2-4,8-14}, the formation of various types of heterojunctions (type-I, II, and III) is limited by threading dislocations, which are normally caused at the junction interface by lattice mismatch during film growth. the heterojunction interfaces suffered significantly from defects, such as dislocation by lattice mismatch. These imperfections at the interface degrade the NDR device performance as the carriers are trapped and recombined at the defect sites. Although the threading dislocation that increases the valley current of the NDR device can be reduced by applying superlattice and nanowire structures, it is hard to avoid that the fabrication process becomes more complex.”~~

~~“Fig. 2b shows the current–voltage (*I*-*V*) characteristic of the NDR device on a linear scale. Here, the NDR behavior was observed between 0.4 V and 0.9 V with a PVCR of 4.2, which is the highest value in previously reported NDR devices based on 2D materials^{5-7,18,19} and is also comparable to the values for other NDR devices fabricated on conventional 3D semiconductors, such as Si, Ge, and III-V compound semiconductors^{2,3,8,10-12}.”~~

(2) In page 4, the claim of " ... electrostatic doping ... " is incorrect. Any useful electronic device will need to be doped to be controlled; it does not help in adjusting threshold voltages etc if the doping is left to nature. This claim is repeated in page 6. This is a fundamental underestimation of the importance of doping in device physics.

ANS: What we wanted to say in this sentence is that a highly doped n^+/p^+ tunnel heterojunction can be implemented without using any doping process (electrostatic doping by gate bias, chemical doping process, etc.). This does not mean that the doping process is not important in device physics, but rather that the fabrication process for the tunnel heterojunction of the NDR device becomes very simple. As already mentioned, the highly doped n^+/p^+ tunnel heterojunction could easily be implemented by forming a type-III broken-gap band alignment.

(3) Because the thickness of the BP layer is so high, how is the gating affecting the conductance? If it is changing the conductivity of the entire channel, it should reach in and also affect the band alignments, etc. This is not included in the analysis or model.

ANS: Because of the large thickness of the BP layer, the top region of the BP film is expected to be weakly modulated by gate bias (FYI, this top BP region faces ReS₂ to form the type-III junction). If the conductivity of the top BP region is unaffected by gate bias, the drain current of the BP TFT (the solid line in the figure below) will be probably unchanged (or only slightly changed) as a function of gate voltage (like the red dashed line) because the current flowing through the BP bulk region dominates the drain current at the off-state. For reference, the deterioration of the on/off-current ratio in BP TFTs with increasing BP film thickness was also reported^{R.4}.

[R.4] L. Li *et al.* *ACS Nano* **10**, 4672-4677 (2016).

As shown in the figure below, we modified the analytic NDR device model by considering the Fermi level modulation *via* the gating effect. However, it was hard to accurately anticipate the gate e-field effect on the top region in the BP film due to the screening effect of the bottom region. Thus, as shown below, we estimated the gating effect in the BP/ReS₂ NDR device by assuming that the Fermi level of the BP region shifts downward by 0.1 eV with decreasing gate voltage, from 30 V to -30 V, in the analytic model (0.1 eV per 60 V = 0.00167 eV/V).

In order to clarify this, we added the following sentence in the article.

“The reduction of peak-current in BP/ReS₂ NDR devices with decreasing gate voltage could also be estimated using the I_D - V_D curves calculated by the analytic model (Supporting Information Fig. S5).”

We also modified Supporting Information Figure S5 and added the related sentences.

In the figure caption:

“(b) Theoretically calculated I_D - V_D curves of the BP/ReS₂ NDR device as gate voltage decreases from 30 V to -30 V.”

In the Supporting Information:

“As shown in Figure S5b, the reduction of peak-current in BP/ReS₂ NDR devices with decreasing gate voltage could also be estimated using the I_D - V_D curves calculated by the analytic model. In the

analytic NDR device model, Fermi level modulation of the BP region *via* the gating effect was considered by assuming that the Fermi level shifts downward by 0.1 eV with decreasing gate voltage from 30V to -30 V (0.1 eV per 60 V = 0.00167 eV/V).”

(4) Page 6: PVCRs exceeding the reported values here by orders of magnitude have been observed in 3D semiconductor tunnel junctions.

ANS: As already addressed in Q/A #1, we agree with the reviewer comment that the NDR devices based on SiGe or III-V heterojunctions exhibited high PVCR values (above 10) at room temperature and we corrected the related sentences (see above Q/A #1).

However, PVCR values (between 3 and 5) in our NDR device are sufficient for typical logic applications^{R.5,6}. Although a relatively lower PVCR value (4.2 at room temperature) was achieved in this BP/ReS₂ heterojunction-based NDR device compared to the values of SiGe or III-V semiconductor-based NDR devices, our NDR device is still a good candidate for realizing MVL circuits.

[R.5] J. M. Martínez-Duart, R. J. Martín-Palma, and F. Agulló-Rueda, *Nanotechnology for Microelectronics and Optoelectronics* Elsevier, Amsterdam (2006).

[R.6] J. P. Sun *et al. Proceedings of the IEEE* **86**, 641-661 (1998).

(5) The modeling in the supplemental material must provide the parameters that were used, they are missing.

ANS: As the reviewer suggested, we tabulated the parameters used in the proposed analytic NDR device model in Supporting Information Table S1.

Used material parameters for the analytic NDR device model

$E_{V, BP}$ [eV]	$E_{C, ReS2}$ [eV]	$E_{F, BP}$ [eV]	$E_{F, ReS2}$ [eV]	$E_{g, BP}$ [eV]	$E_{g, ReS2}$ [eV]
-----------------------	---------------------	-----------------------	---------------------	-----------------------

4.59*	4.68*	4.5	5.1	0.39*	1.37*
m_{BP}^* [kg]	m_{ReS2}^* [kg]	η_{id}	α	I_0 [A]	
$7.56 \times 10^{-31}^*$	$2.71 \times 10^{-31}^*$	12 @ RT	1.27	2.28×10^{-10} @ RT	

*The material parameters were obtained from those previously reported in the literature^{S4-S8}.

Supplementary Table S1. The parameters used for the analytic NDR device model.

We also added related references in Supporting Information.

S4. Perello, D. J., Chae, S. H., Song, S. & Lee, Y. H. High-performance n-type black phosphorus transistors with type control via thickness and contact-metal engineering. *Nat. Commun.* **6**, 7809 (2015).

S5. Liu, X. *et al.* Black Phosphorus Based Field Effect Transistors with Simultaneously Achieved Near Ideal Subthreshold Swing and High Hole Mobility at Room Temperature. *Sci. Rep.* **6**, 24920 (2016).

S6. Ho, C. H., Huang, Y. S., Chen, J. L., Dann, T. E. & Tiong, K. K. Electronic structure of ReS₂ and ReSe₂ from first-principles calculations, photoelectron spectroscopy, and electrolyte electroreflectance. *Phys. Rev. B* **60**, 15766-15771 (1999).

S7. Liu, H. *et al.* Phosphorene: An Unexplored 2D Semiconductor with a High Hole Mobility. *ACS Nano* **8**, 4033-4041 (2014).

S8. Yu, Z. G., Cai, Y. & Zhang, Y.-W. Robust Direct Bandgap Characteristics of One- and Two-Dimensional ReS₂. *Sci. Rep.* **5**, 13783 (2015).

We've also modified the following sentence in the article.

“The equations related to the current transport mechanisms can be found in Supporting Information Fig. S4 and the parameters used in the analytic model are tabulated in Supporting Information Table S1.”

REVIEWERS' COMMENTS:

Reviewer #1 (Remarks to the Author):

The authors have addressed all the previous comments from the reviewer. One minor note is that the authors updated Fig. 1g, but not Fig. 1h and 2d regarding the band offset at the junction. Please correct this discrepancy before publication. The reviewer otherwise thinks that this manuscript is ready to be published in Nature Communications.

Reviewer #2 (Remarks to the Author):

The author has revised the manuscript properly based on the comments that I gave last time. There seems no critical problems left in terms of logical explanation on the experimental data. I believe this paper contains meaningful work which will be helpful for other researchers in the same field. Therefore, I wish this paper to be published in Nature communications.

Point by Point Response to the Reviewers

Reviewer: 1 Comments to the Author

The authors have addressed all the previous comments from the reviewer.

We would like to thank you for reviewing our paper; we appreciate your insightful comments on our research. We have revised the manuscript according to your suggestions and believe that the revisions have improved the paper.

Please find below our responses (in blue) to each specific comment (in black) provided by the reviewer. In addition, revisions to the original article are highlighted in red.

(1) One minor note is that the authors updated Fig. 1g, but not Fig. 1h and 2d regarding the band offset at the junction. Please correct this discrepancy before publication. The reviewer otherwise thinks that this manuscript is ready to be published in Nature Communications.

ANS: We modified Figure 1h and 2d as below.

In Figure 1h:

In Figure 2d:

Reviewer: 2 Comments to the Author

The author has revised the manuscript properly based on the comments that I gave last time. There seems no critical problems left in terms of logical explanation on the experimental data. I believe this paper contains meaningful work which will be helpful for other researchers in the same field. Therefore, I wish this paper to be published in Nature communications.

We would like to thank you for reviewing our paper; we appreciate your insightful comments on our research.